# Metal-organic framework boosts heterogeneous electron donor–acceptor catalysis

Jiaxin Lin[1,2], Jing Ouyang[1,2], Tianyu Liu[1], Fengxing Li[1], Herman Ho-Yung Sung [1], Ian Williams [1] & Yangjian Quan [1] ✉

Metal-organic framework (MOF) is a class of porous materials providing an excellent platform for engineering heterogeneous catalysis. We herein report the design of MOF Zr-PZDB consisting of $Zr_6$-clusters and PZDB (PZDB = 4,4'-(phenazine-5,10-diyl)dibenzoate) linkers, which served as the heterogeneous donor catalyst for enhanced electron donor–acceptor (EDA) photoactivation. The high local concentration of dihydrophenazine active centers in Zr-PZDB can promote the EDA interaction, therefore resulting in superior catalytic performance over homogeneous counterparts. The crowded environment of Zr-PZDB can protect the dihydrophenazine active center from being attacked by radical species. Zr-PZDB efficiently catalyzes the Minisci-type reaction of N-heterocycles with a series of C-H coupling partners, including ethers, alcohols, non-activated alkanes, amides, and aldehydes. Zr-PZDB also enables the coupling reaction of aryl sulfonium salts with heterocycles. The catalytic activity of Zr-PZDB extends to late-stage functionalization of bioactive and drug molecules, including Nikethamide, Admiral, and Myristyl Nicotinate. Systematical spectroscopy study and analysis support the EDA interaction between Zr-PZDB and pyridinium salt or aryl sulfonium salt, respectively. Photoactivation of the MOF-based EDA adduct triggers an intra-complex single electron transfer from donor to acceptor, giving open-shell radical species for cross-coupling reactions. This research represents the first example of MOF-enabled heterogeneous EDA photoactivation.

Organic photosensitizers, a representative class of non-metal photocatalysts, have received increasing research interest[1–3]. In fact, nature evolves pigments to utilize solar energy for important chemical transformations[4]. Compared with inorganic photocatalysts (PCs), the organic ones possess a relatively broader redox window, and thus a stronger capability to active inert molecules[3]. However, their disadvantages of a shorter excited-state lifetime and lower stability lead to an inferior catalytic performance in some cases. For example, diaryl dihydrophenazines are visible-light PCs with strongly reducing ability ($E^0(PC^{•+}/^3PC^*) < -2$ V versus saturated calomel electrode) upon excitation. Although they efficiently catalyzed the atom transfer radical

polymerization[5], the carbon radical intermediate was recently found to attack the PCs (Fig. 1a)[6,7]. The susceptibility of dihydrophenazine PCs to the reaction with open-shell radical species may restrict their applications in photoredox catalysis.

In addition to traditional photoredox catalysis enabled by photosensitizers, the recently developed electron donor–acceptor (EDA) photoactivation provides an alternative mode for photochemical synthesis[8,9]. The proper electron donor catalyst can interact with electron acceptor substrates to form a photoactive EDA complex. Upon light irradiation, intra-complex single electron transfer (SET) from donor to acceptor occurs to give active open-shell species for

[1]Department of Chemistry and the Hong Kong Branch of Chinese National Engineering Research Centre for Tissue Restoration & Reconstruction, The Hong Kong University of Science and Technology (HKUST), Kowloon, Hong Kong SAR, China. [2]These authors contributed equally: Jiaxin Lin, Jing Ouyang. ✉e-mail: chyjquan@ust.hk

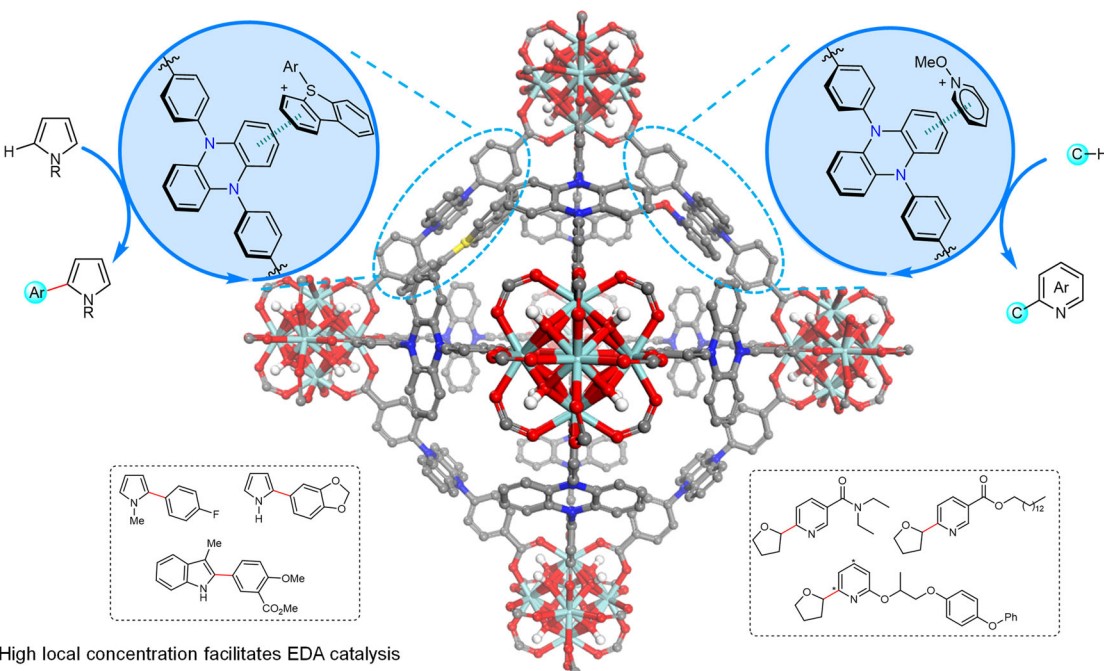

**Fig. 1 | Diaryl dihydrophenazine catalysis. a** Representative example of visible-light dihydrophenazine catalysis and its limitation. Ar aromatics, SCE saturated calomel electrode. **b** Enhancing the performance of dihydrophenazine EDA photoactivation by MOF platform. EDA electron donor–acceptor, MOF metal-organic framework; light blue: Zr, red: O, blue: N, gray: C.

further transformations[10–13]. In most cases, the EDA interaction is "weak", thus a relatively high concentration is preferred to promote the chances of generating EDA adducts. Despite fruitful achievements, the development of the corresponding heterogeneous version of EDA photoactivation remains inaccessible.

Metal-organic frameworks (MOFs)[14–26], a class of 3D porous materials by bridging metal-containing second building units (SBUs) with organic linkers, have proved their superiority in engineering heterogeneous catalysis due to the wide synthetic tunability and unique porous structure[27–43]. Specific to MOF-based photocatalysis, factors including site isolation effect, confinement effect, high local concentration, and active-center protection endow MOF catalysts with superior catalytic performance compared with homogeneous counterparts[44–58]. A variety of metal-based PCs have been integrated into MOFs for photoinduced synthesis, while the merger of organic PCs remains less investigated because of their typically low symmetry and high steric demand[59–62]. Among the aforementioned factors, a high local concentration of active centers in MOF was surmised to promote the EDA interaction and related EDA photoactivation, which remains elusive in heterogeneous catalysis. Moreover, integrating diaryl dihydrophenazine into MOFs would reduce the risk of being attacked by radical intermediates and therefore expand their catalytic applications. Herein, we report the design of photoactive MOF Zr-PZDB (PZDB = 4,4'-(phenazine-5,10-diyl)dibenzoate), consisting of $Zr_6$-SBUs and PZDB connecting ligands (Fig. 1b). Upon visible light irradiation, Zr-PZDB competently catalyzed the Minisci-type cross-coupling of *N*-heterocycles with ethers, alcohols, non-activated alkanes, amides, and aldehydes. Zr-PZDB also enabled the coupling reaction of aryl sulfonium salts with heterocycles. Furthermore, the late-stage functionalization of complex drug or bioactive molecules was attained by using

Zr-PZDB as the catalyst[63]. In contrast, the homogeneous counterparts PZDB-H or PZDB-Me exhibited inferior catalytic efficiency. Our systematical spectroscopy study and analysis revealed the enhanced EDA interaction between Zr-PZDB and pyridinium salt or sulfonium salt. The subsequent photoinduced intra-complex SET allowed the generation of radicals and enabled the corresponding coupling reactions.

## Results and discussion

The PZDB-H linker was synthesized according to the reported procedure (see Supplementary Information for details)[64]. PZDB-H presents a rigid single-crystal structure (Supplementary Fig. S3), suggesting its potential to be bridging ligands for MOF synthesis. Solvothermal reaction of PZDB-H ligand with $ZrCl_4$ in $N,N$-dimethylformamide (DMF) using trifluoroacetic acid as the modulator delivers octahedron single crystals of Zr-PZDB (Fig. 2a and Supplementary Fig. S4a; during the preparation of this manuscript, the Queen group reported a convenient synthesis of a series of MOFs with piperazine core[65]. The single-crystal structure suggests a formula of $Zr_6(\mu_3\text{-O})_4(\mu_3\text{-OH})_4(PZDB)_6$ for Zr-PZDB, which was further verified by TGA (thermogravimetric analysis, Supplementary Fig. S8) result with the residue weight of 23.7 wt% for $ZrO_2$ close to the theoretical value of 23.1 wt% from the formula. Zr-PZDB exhibits a space group of Fm−3m (No. 225; see Supplementary Table S2) and fcu topology, wherein $Zr_6O_4(OH)_4$ clusters are connected by 12 PZDB linkers in the face-centered-cubic array, in line with the iso-reticular structures of UiO-68[66]. The dihydrophenazine moiety in PZDB of Zr-PZDB retains the planar geometry. However, it is disordered due to the rotation around the PZDB backbone (Supplementary Fig. S4b). The shortest distance between two adjacent linkers is measured as ~2 Å, indicating the relatively steric demanding environment (Supplementary Fig. S5), which might account for the excellent protection of PZDB active centers by Zr-PZDB (vide infra).

Powder X-ray diffraction (PXRD) analyses indicated the high phase purity of Zr-PZDB. The corresponding PXRD pattern matches well with the simulated one from the single crystal structure (Fig. 2b). In addition, Zr-PZDB retained the crystalline structure after being soaked in a series of solvents, as evidenced by PXRD analyses (Fig. 2b and Supplementary Fig. S10). Scanning electron microscopy (SEM) imaging showed the octahedron morphology of Zr-PZDB (Fig. 2c). High-resolution transmission electron microscopy (HRTEM) imaging of Zr-PZDB revealed a distance of ~2.4 nm between two $Zr_6$-SBUs (Fig. 2d), matching well with the distance of ~2.3 nm in the single crystal structure (Supplementary Fig. S6). IR spectra of PZDB-H and Zr-PZDB were collected. A stretching band at 1690 $cm^{-1}$ assignable to the carbonyl group was observed for PZDB-H, while the carbonyl groups in Zr-PZDB showed the stretching band at 1654 $cm^{-1}$ (Fig. 2e). The difference originated from the bonding of carboxylic group with metals. Only one set of signals assigned to PZDB-H was detected by $^1$H NMR analysis of the digested Zr-PZDB, indicating the inertness of the bridging ligand under MOF preparation conditions and the high purity of MOF (Fig. 2f). The porosity of Zr-PZDB was investigated by gas absorption analyses (Supplementary Fig. S9). The Brunauer−Emmett−Teller surface area of Zr-PZDB was measured as 715 $m^2$/g with the calculated main pore window of ~1.3 nm. As shown in Fig. 2g, upon excitation at 398 nm, PZDB-H exhibited two characteristic emission peaks at 450 and 550 nm, respectively. However, the intensity of the 450 nm peak was significantly decreased, with a main emission peak at 550 nm for Zr-PZDB. Meanwhile, PZDB-H and Zr-PZDB showed similar excitation signals for the emission at 550 nm.

Functionalization of $N$-heterocycles is of great importance from their potential bioactivity viewpoints[67]. In addition to traditional synthetic routes, photoinduced derivatization via the intermediacy of open-shell radicals represents an alternative strategy. Taking this into consideration, the Minisci-type cross-coupling of pyridinium salt[68–70] with tetrahydrofuran was chosen as the model reaction to evaluate the

catalytic performance of Zr-PZDB. As shown in Table 1, Zr-PZDB competently catalyzed the cross-coupling reaction upon blue LED irradiation (Kessil PR160L-427, 390−470 nm, Supplementary Fig. S30) to give the target product 3a in 80% isolated yield. Screening other solvents proved that $CH_3CN$ was the optimal choice (entries 2−4, Table 1). Base $NaHCO_3$ outperformed other inorganic bases, due probably to its relatively better solubility (entries 5,6, and 8, Table 1). The use of $NEt_3$ led to a reduced yield of 12% for 3a (entries 5-8, Table 1), because of its potential to quench methoxy radical via hydrogen atom transfer reaction. The linker PZDB-H or PZDB-Me instead of Zr-PZDB as the catalyst delivered 3a in only 6% and 13% yields, respectively (entries 9 and 10, Table 1). Loading PZDB-H into activated carbons as the catalyst provided a yield of 15% for 3a (entry 17, Table 1 and Supplementary Fig. S36). These results indicated the important role of the MOF platform in the catalytic performance. Noteworthily, the use of several representative molecular photosensitizers, including $Ir(ppy)_3$, $Ru(bpy)_3^{2+}$, and Eosin Y, afforded inferior results (entries 11−13, Table 1). The control experiments suggested that both light and MOF catalyst were indispensable for the efficient transformation (entries 14 and 15, Table 1). A combination of $ZrCl_4$ and PZDB-H instead of Zr-PZDB led to trace amounts of 3a (entry 16, Table 1).

The individual solution of 1a and suspension of Zr-PZDB were almost colorless; however, their mixture exhibited a light-yellow color (Fig. 2h). This phenomenon suggested the potential involvement of EDA interaction. Noteworthily, the Lakhdar lab pioneered the identification of an EDA interaction between the organic dye Eosin Y and $N$-ethoxy-2-methylpyridinium tetrafluoroborate[71]. To further verify the feasibility, a detailed spectroscopy study was then carried out. UV-vis spectrum of the mixture presented a significantly enhanced peak from 330 to 430 nm with respect to the absorption profiles of the individual 1a and Zr-PZDB (Fig. 2h), indicating the formation of an EDA complex. In sharp contrast, the UV-vis spectrum of a mixture of 1a and PZDB-H was almost identical to the sum of individual spectra of 1a and PZDB-H (Fig. 2i). The obvious difference might be attributed to the high local concentration of dihydrophenazine active centers in Zr-PZDB, which are believed to benefit the formation of EDA adducts kinetically. UV-vis spectra of a series of mixtures with different Zr-PZDB/1a ratios were subsequently collected (the ratio of Zr-PZDB is based on the PZDB linker, Fig. 2j). The maximum absorption enhancement was observed for a Zr-PZDB/1a ratio of 1/1 (green line in Fig. 2j). The corresponding Job plot further verified the optimal ratio of 1/1 for the plausible EDA complex (Fig. 2k and Supplementary Table S3). In addition, no absorption enhancement was detected by mixing 1a with $NaHCO_3$ (Supplementary Fig. S13). To further illustrate the role of EDA interaction in this photoinduced Minisci-type transformation, a 420 nm band filter, which blocks lights with a wavelength smaller than 420 nm, was used with a Kessil PR160L-427 lamp (390-470 nm). No product 3a was detected with 1a being intact (Supplementary Figs. S32 and S33). Furthermore, the utilization of the Kessil PR160L-390 lamp (370-420 nm) provided a yield of ~50% for 3a (Supplementary Fig. S32), with the detection of ~50% 4-methylpyrdine (Supplementary Fig. S34). The addition of a 380 nm band filter increased the yield to ~80% (Supplementary Figs. S32 and S35). These results indicate that (1) light with a wavelength larger than 420 nm cannot induce the reaction; (2) light less than 380 nm in wavelength might decrease the selectivity for 3a; (3) the likely involvement of EDA interaction in this photoinduced transformation[72]. To demonstrate the considerable interaction between Zr-PZDB and pyridinium salt 1a, an adsorption control experiment was designed and performed (Supplementary Fig. S38). After being stirred with 1 equiv of Zr-PZDB overnight, the $CH_3CN$ solution of 1a only retained 5% of pyridinium salt. In other words, Zr-PZDB competently adsorbed 95% of 1a, indicating their strong interaction.

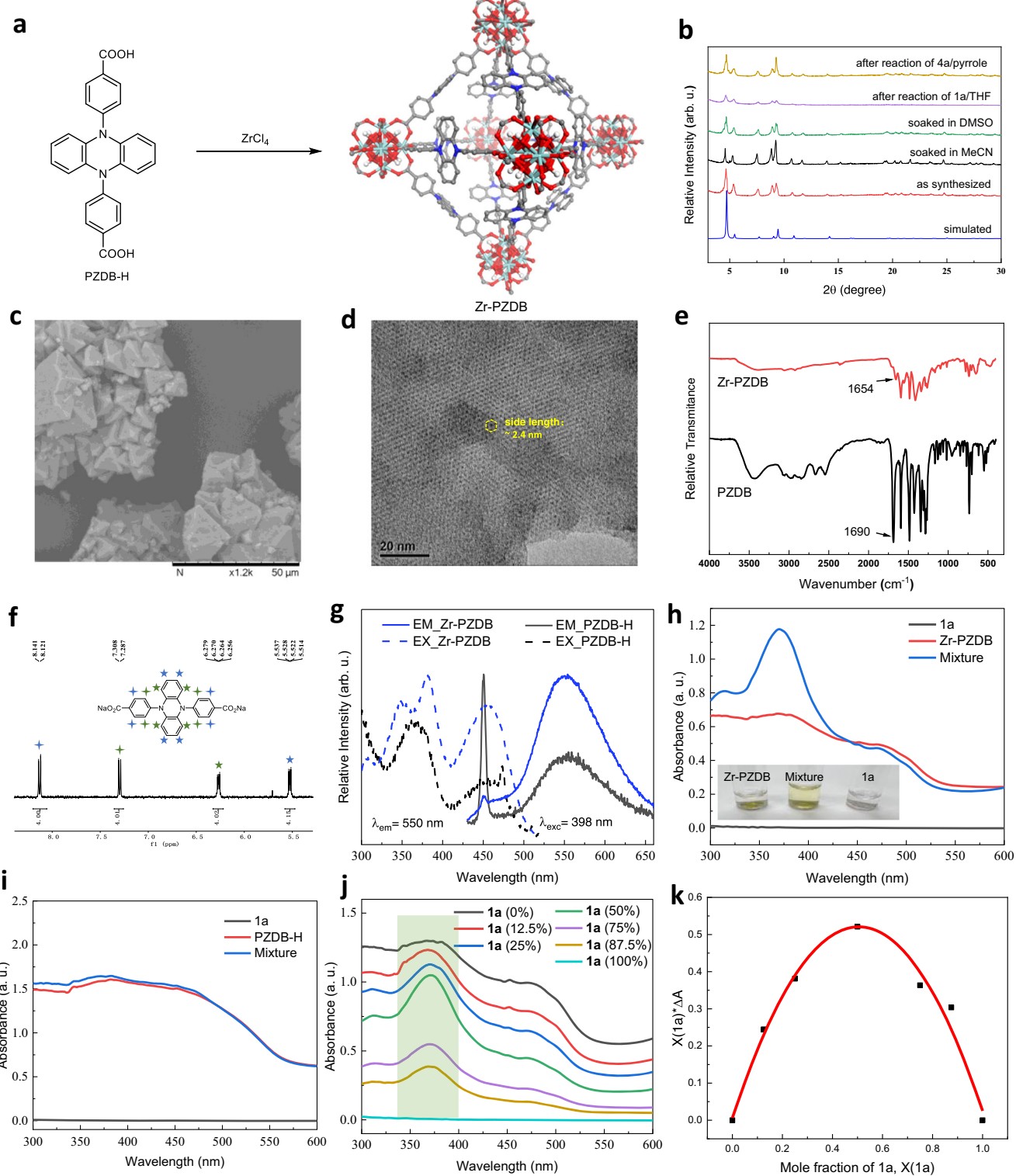

**Fig. 2 | Synthesis and characterization of Zr-PZDB. a** Schematic showing the synthesis of Zr-PZDB and the structure: X-ray single-crystal structure represented by an octahedron-like cage; disorder and H atoms are omitted for clarity; light blue: Zr, red: O, blue: N, gray: C. **b** PXRD patterns of Zr-PZDB; arb. u. = arbitrary units. **c**, **d** SEM (**c**) and HRTEM (**d**) of Zr-PZDB. **e** IR spectra of Zr-PZDB and PZDB-H. **f** Zoom-in $^{1}$H NMR of digested Zr-PZDB in DMSO-$d_6$. **g** Normalized excitation and emission spectra of Zr-PZDB and PZDB-H in MeCN; arb. u. = arbitrary units. **h** UV-vis spectra of **1a**, Zr-PZDB, and their mixture in MeCN ($2.5 \times 10^{-3}$ M); a. u. = absorbance units. **i** UV-vis spectra of **1a**, PZDB-H, and their mixture in MeCN ($2.5 \times 10^{-3}$ M); a. u. = absorbance units. **j** UV-vis spectra of mixtures of **1a** and Zr-PZDB with different ratios of **1a** in MeCN ($5 \times 10^{-3}$ M); a. u. = absorbance units. **k** Job's plot based on UV-vis data in (**j**).

## Table 1 | Optimization of reaction conditions[a]

| Entry | Variations from "standard conditions" | Yield (%) |
|---|---|---|
| 1 | No variation | 80 |
| 2 | DCE instead of CH$_3$CN | 66 |
| 3 | DCM instead of CH$_3$CN | 42 |
| 4 | Toluene instead of CH$_3$CN | 56 |
| 5 | Na$_2$CO$_3$ instead of NaHCO$_3$ | 11 |
| 6 | K$_2$CO$_3$ instead of NaHCO$_3$ | 35 |
| 7 | Et$_3$N instead of NaHCO$_3$ | 12 |
| 8 | Cs$_2$CO$_3$ instead of NaHCO$_3$ | 21 |
| 9 | PZDB-Me instead of Zr-PZDB | 13 |
| 10 | PZDB-H instead of Zr-PZDB | 6 |
| 11 | Ir(ppy)$_3$ instead of Zr-PZDB | 8 |
| 12 | Ru(bpy)$_3^{2+}$ instead of Zr-PZDB | 41 |
| 13 | Eosin Y instead of Zr-PZDB | 70 |
| 14 | Without light | Trace |
| 15 | Without Zr-PZDB | Trace |
| 16 | ZrCl$_4$ and PZDB-H instead of Zr-PZDB | Trace |
| 17 | PZDB-H@C instead of Zr-PZDB | 15 |

[a]Standard conditions: pyridinium salt (0.05 mmol), NaHCO$_3$ (0.1 mmol), Zr-PZDB (2.5 μmol, 5 mol % based on the linker), THF (3.8 mmol, 0.3 mL), CH$_3$CN (0.5 mL), N$_2$, r.t., PR160L-427 (390–470 nm), 24 h; red: the newly formed bond; yields of isolated **3a**; ppy = 2-phenylpyridine, bpy = 2,2′-bipyridine, @C = loaded into activated carbons.

The scope of dehydrogenative cross-coupling was then explored, and the results were compiled in Fig. 3. With Zr-PZDB as the catalyst, five different kinds of C-H coupling partners, including ethers, alcohols, unactivated alkyls, amides, and aldehydes, reacted with pyridinium salts efficiently, evidencing the versatility of MOF catalyst. Both cyclic and linear ethers worked well to give products **3a**-**3d** in moderate to very good yields. Pyridinium salts bearing various substituents such as alkyl, phenyl, ester, cyanide, methoxy, and trifluoromethyl groups were compatible (**3e**-**3k** and **3m**-**3n**), and no obvious electronic effects were observed. The substrate 1-methoxylepidinium methyl sulfate afforded the corresponding **3 l** in 70% yield. Pyridinium salts without a substituent on the C$_4$ position delivered *ortho*- and *para*-functionalized products with C$_2$ to C$_4$ ratios of 1.2 to 2.9 (**3o**-**3s**). Noteworthily, relatively bulky substituents on the C$_3$ position sterically protect the adjacent C$_2$ and C$_4$ positions, resulting in very good C$_6$ selectivity (**3aj** and **3al**). Methanol and ethanol also served as good coupling partners, and the corresponding pyridinyl alcohols **3t**-**3v** were obtained. Without C$_4$ protection, both *ortho*- and *para*-substituted products were isolated with ratios from 1.4 to 5.4 (**3w**-**3y**).

Direct replacement of ether by amine, for example, triethylamine, did not yield the coupling product. The easy oxidation of the corresponding α-C radical of triethylamine might account for the failed attempt. However, the dimethylacetamide (DMA) was tolerated to produce **3ad**-**3ag** efficiently. Furthermore, the non-activated, cyclic alkanes worked as appropriate feedstocks, giving the resultant **3z**-**3ac** in good yields. The ring size had little effect on the reaction efficiency. In addition to nucleophilic C(sp$^3$)-H coupling partners, substrates

containing carbonyl C(sp$^2$)-H also underwent the cross-coupling reaction. Treatment of dimethylformamide (DMF) or benzaldehyde with 1-methoxylepidinium methyl sulfate under standard reaction conditions afforded **3ad** and **3ae**, respectively. No target product was observed when **1a** was used as the substrate. Noteworthily, Zr-PZDB effectively catalyzed the late-stage functionalization of complicated drug or bioactive molecules containing Nikethamide (**3aj**), Admiral (**3ak**), and Myristyl Nicotinate (**3al**).

To further evaluate the ability of Zr-PZDB donor catalyst, aryl sulfonium salt was examined as the acceptor and substrate (Fig. 4). Upon getting one electron, aryl sulfonium salt would undergo decomposition to generate aryl radical, which is more reactive and sometimes incompatible with homogenous catalytic systems[13,73]. To our delight, Zr-PZDB effectively catalyzed the coupling reaction between aryl sulfonium salt **4a** and 1-methylpyrrole upon light (PR160L-427) irradiation, giving **6a** in 87% isolated yield (entry 1 in Supplementary Table S4). Replacement of Zr-PZDB by PZDB-H or PZDB-Me led to decreased yields of 39% and 12% for **6a**, respectively (entries 2 and 3 in Supplementary Table S4). PZDB-H@C as the catalyst provided a reduced yield of 36% (entry 7 in Supplementary Table S4 and Supplementary Fig. S37). The combination of ZrCl$_4$ and PZDB-H instead of Zr-PZDB afforded **6a** in 45% yield (entry 4 in Supplementary Table S4). The absence of Zr-PZDB or light resulted in almost no target product (entries 5 and 6 in Supplementary Table S4), suggesting their importance in the coupling reaction.

A series of aryl sulfonium salts and heterocycles were then evaluated (Fig. 4). Functional groups, including alkyls, ethers, esters, halides, and triflate, were well tolerated, among which esters, halides, and triflate could be conveniently transformed to other substituents (**6a**-**6h**). Pyrrole, indole, thiophene, and furan underwent the coupling reaction smoothly to deliver **6i**-**6r** in 52-91% isolated yields. No obvious electronic effects were observed. In view of the prevalence of heterocycle derivatives in drug or bioactive molecules, this methodology might find potential applications in biochemistry.

Investigation on the EDA interaction between Zr-PZDB and **4a** was also performed. UV-vis spectrum of the corresponding mixture showed the absorption enhancement from 350 nm to 430 nm compared with the absorption profiles of the individual **4a** and Zr-PZDB (Fig. 5a). In contrast, no obvious absorption enhancement was observed from the UV-vis spectrum of the mixture of PZDB-H and **4a** (Fig. 5b). Moreover, at a high concentration ($2.5 \times 10^{-3}$ M), absorption enhancement was more obvious for the mixture of Zr-PZDB/**4a** with little changes for the mixture of PZDB-H/**4a** (Supplementary Fig. S12). These results further evidenced the enhancement effect of the MOF platform on the EDA interaction. To verify the involvement of radical species, radical capture and clock experiments were carried out (Fig. 5c). The addition of a radical scavenger, TEMPO, to the reaction mixture of **1a**/THF completely inhibited the formation of product **3a**, with the detection of radical capture species **7** by high-resolution mass spectrometry (HRMS). On the other hand, radical scavenger 1,1-diphenylethene successfully captured the more reactive aryl radical as proved by the formation of **8**. In addition, the results of radical clock experiments further verified the involvement of alkyl and aryl radicals in the coupling reactions, respectively.

Light on/off experiments for the coupling between **1a** and THF indicated the continuation of the reaction in the dark (Fig. 5d), suggesting a radical chain pathway. In sharp contrast, almost no reaction was detected for coupling between **4a** and pyrrole in the dark (Fig. 5e). In addition, its corresponding quantum yield was measured as 0.003 (Supplementary Information "Determination of quantum yield" section), negating the radical progression pathway[74]. A similar phenomenon was observed in hot-filtration control experiments. The removal of the Zr-PZDB catalyst from the reaction mixture of **1a** and THF after 12-hour irradiation did not totally shut down the reaction. The yield for **3a** was increased from 36% to 52% without MOF catalyst under light

**Fig. 3 | Zr-PZDB catalyzed Minisci-type cross-coupling reactions.** Reactions were conducted at 0.05 mmol scale; yields of isolated products; red: the newly formed bond; blue: the reaction sites.

irradiation (Supplementary Fig. S26), albeit a higher yield of 80% in the presence of MOF catalyst. However, little yield increase for **6a** was observed after removing the Zr-PZDB catalyst from the reaction mixture of **4a** and pyrrole (Supplementary Fig. S27). In addition, the leaching of Zr was detected as <0.3% for both coupling reactions by ICP-MS, demonstrating the stability of Zr-PZDB under the coupling reaction conditions. These results combined with the light on/off experiments indicated: (1) the coupling reaction between **1a** and THF would involve the radical chain pathway, whereas the reaction of **4a** and pyrrole did not undergo radical progression; and (2) the heterogeneous nature of Zr-PZDB catalysis in both coupling reactions.

The Zr-PZDB catalyst was recovered and used in three runs of cross-coupling reactions of **1a**/THF and **4a**/pyrrole with slightly decreased catalytic performance (Supplementary Figs. S16 and S17).

Zr-PZDB retained the crystalline structure, as evidenced by the matching PXRD pattern after the reaction (Fig. 2b). To compare the stability of PZDB active center in Zr-PZDB and homogeneous counterpart, attempts to recover and characterize the catalysts after the reaction were carried out. [1]H NMR spectrum of the digested Zr-PZDB-AR (AR = after reaction) was almost identical to that of the as-prepared Zr-PZDB (Fig. 5g and Supplementary Fig. S18). In contrast, thin layer chromatography (TLC) and [1]H NMR analyses of the homogeneous reaction mixture indicated the disappearance of the PZDB-Me catalyst (Supplementary Figs. S19 and S20). HRMS analysis further suggested the formation of mono-, di-, and tri-substituted derivatives of PZDB-Me probably originating from the attack of the corresponding aryl radicals (Fig. 5f and Supplementary Fig. S21). These results verified the reported susceptibility of dihydrophenazine molecular catalyst against

**Fig. 4 | Zr-PZDB catalyzed cross-coupling of aryl sulfonium salts and heterocycles.** Reactions were conducted at 0.2 mmol scale; yields of isolated products; red: the newly formed bond.

active radical species and highlighted the protection ability of the MOF platform.

On the basis of the above experimental results and seminal literature precedents[75–80], plausible reaction mechanisms are proposed (Fig. 5h, i). For Minisci-type reaction, integration of Zr-PZDB with pyridinium salt gives an EDA complex. Upon visible light irradiation, the intra-complex SET from Zr-PZDB to the pyridinium salt occurs to afford neutral pyridine and methoxy radical. The latter competently abstracts hydrogen from C-H bonds to generate carbon-centered radicals, which then attack the pyridinium salt to give the radical adduct intermediate **B**. Deprotonation of **B** by neutral pyridine or NaHCO$_3$ delivers the intermediate **C**. The intermediate **C** then undergoes decomposition to form the final product and methoxy radical. The oxygen radical can further go through HAT with C-H coupling partners to start another cycle. Alternatively, intermediate **C** may reduce the MOF radical cation to regenerate the MOF catalyst and afford the substituted pyridinium salt **D**, which then forms an EDA adduct with Zr-PZDB followed by light excitation and decomposition to give the final product. For coupling reaction between aryl sulfonium salt and pyrrole, Zr-PZDB interacts with sulfonium salt to form the EDA adduct, which undergoes SET upon light irradiation to give MOF radical cation and aryl radical. The latter attacked pyrrole to deliver intermediate **E**, which is then oxidized by MOF radical cation to regenerate the Zr-PZDB catalyst and give intermediate **F**. Deprotonation of **F** finally delivers the coupling products.

In summary, we have integrated dihydrophenazine into MOF to realize a series of cross-coupling reactions. The Zr-PZDB MOF served as the heterogeneous donor catalyst to interact with the pyridinium and sulfonium acceptors for generating the photoactive EDA complexes, respectively. Photoactivation of the EDA adducts triggered the intra-complex SET and the coupling reactions. The high local concentration of PZDB active centers in Zr-PZDB is believed to promote the EDA interaction kinetically, accounting for the superior catalytic performance of Zr-PZDB over homogeneous counterparts. Zr-PZDB competently catalyzed the dehydrocoupling between pyridinium salts and ethers, alcohols, non-activated alkanes, amides, and aldehydes, leading to a wide variety of *N*-heteroarene derivatives. In addition, Zr-PZDB enabled the synthesis of various aryl heteroarenes from aryl sulfonium salts and heterocycles. Some of the resultant products may have potential bioactivity. Moreover, the developed Zr-PZDB catalysis can be used for attaining the late-stage functionalization of drug and/or bioactive molecules, including Nikethamide, Admiral, and Myristyl Nicotinate. The MOF scaffold effectively protected the dihydrophenazine active center, resulting in relatively low catalyst loading and good durability. This research not only highlights the potential of MOF engineering to address the limitations of organic catalysis but also paves an innovative avenue to green and sustainable MOF-based EDA photoactivation.

## Methods

Preparation of Zr-PZDB: PZDB-H (4.2 mg, 0.01 mmol), ZrCl$_4$ (8.4 mg, 0.036 mmol), and CF$_3$COOH (28.5 mg, 0.25 mmol) were mixed in DMF (0.4 mL) in a closable flask. The mixture was then heated at 120 °C for 48 h. After cooling to room temperature naturally, the yellow crystalline solid was obtained by centrifugation and then sequentially washed with DMF three times. Solvent exchange with benzene (3 × 5 mL, replaced by fresh benzene every 8 h) was then conducted. Then, the

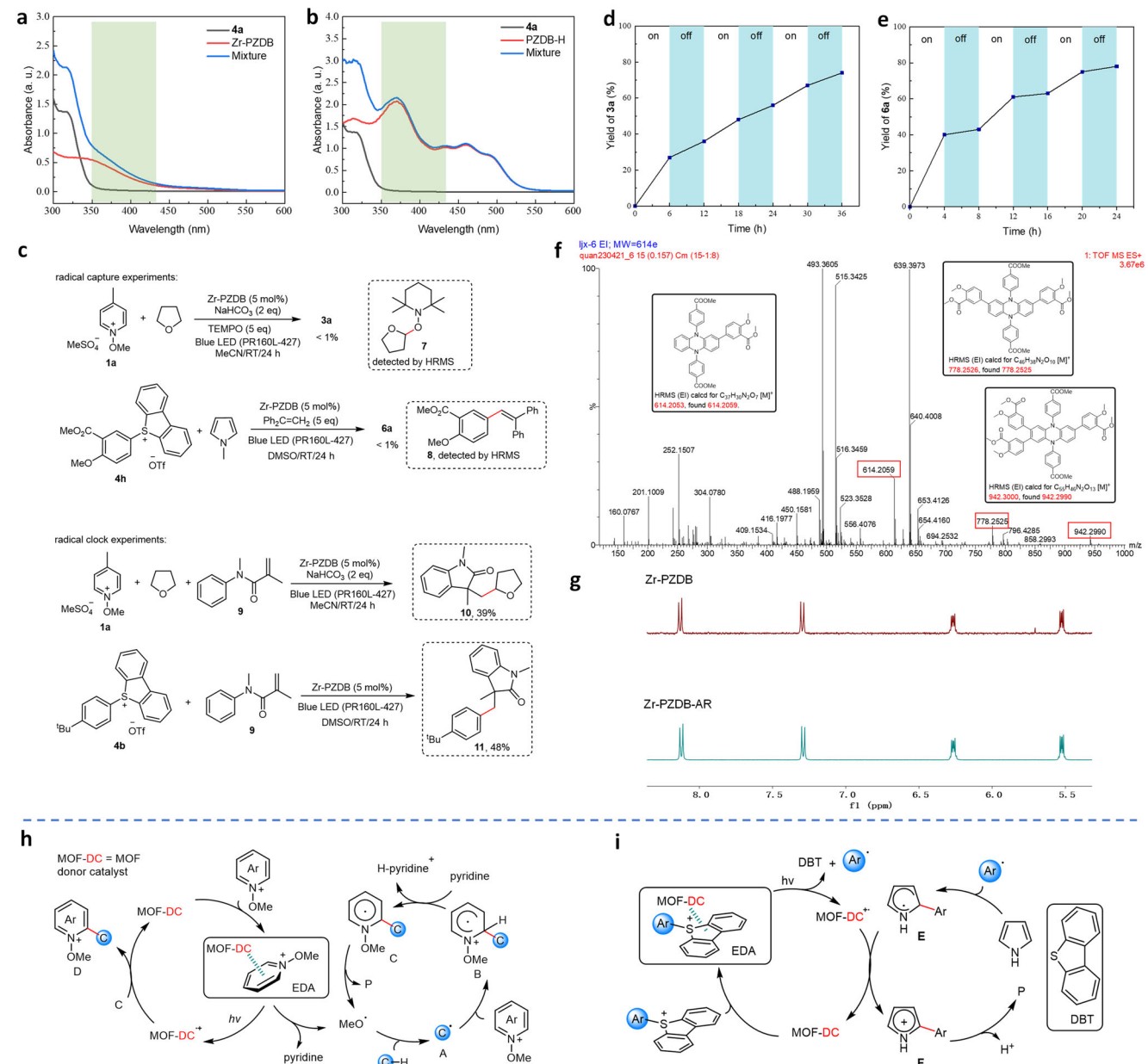

**Fig. 5 | Mechanistic investigation. a** UV-vis spectra of **4a**, Zr-PZDB, and their mixture in DMSO ($5 \times 10^{-4}$ M); a. u. = absorbance units. **b** UV-vis spectra of **4a**, PZDB-H, and their mixture in DMSO ($5 \times 10^{-4}$ M); a. u. = absorbance units. **c** Radical capture and clock experiments. **d**, **e** Light on/off experiments for coupling reactions of **1a**/ THF (**d**) and **4a**/pyrrole (**e**). **f** HRMS spectrum of crude reaction mixture catalyzed by PZDB-Me. **g** Zoom-in $^1$H NMR of digested Zr-PZDB as prepared (up) and after catalytic reaction (down). **h**, **i** Proposed reaction mechanisms for Minisci-type reaction (**h**) and pyrrole arylation (**i**).

resultant crystalline solid was dried by vacuum to afford Zr-PZDB (4.4 mg, 82% yield).

General procedure for Minisci-type reaction: *N*-Methoxy pyridinium methylsulfate (0.05 mmol), CH coupling partners (3.8 mmol), NaHCO$_3$ (8.4 mg, 0.10 mmol), and Zr-PZDB (1.3 mg, 2.5 µmol, 5 mol% based on linker) were mixed in acetonitrile (0.5 mL) in a sealed test tube. The resulting mixture was stirred under blue LED irradiation (PR160L-427, 390-470 nm) at room temperature in a N$_2$ atmosphere for 24 h. After that, the solvent was removed under vacuum, and the residue was subjected to column chromatography on silica gel to give products **3**.

General procedure for heterocycle arylation: Dibenzothiophenium salt (0.2 mmol), heterocycle (8.0 mmol), and Zr-PZDB (5.3 mg, 10.0 µmol, 5.0 mol% based on the linker) were mixed in DMSO (1.0 mL) in a sealed test tube. The resulting mixture was stirred under blue LED

irradiation (PR160L-427, 390–470 nm) at room temperature in a N$_2$ atmosphere for 24 h. After that, the reaction was quenched with aqueous saturated NaHCO$_3$ and diluted with EtOAc. The organic layer was washed with brine, dried with Na$_2$SO$_4$, filtered, and concentrated in vacuo. The residue was subjected to column chromatography on silica gel to give products **6**.

## Data availability
The data supporting the findings of this study are available within the article and its Supplementary Information.

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

## Acknowledgements

We thank Dr. Mengyue Wang and Prof. Dr. Cen Tang for TEM imaging. The authors acknowledge the start-up funding (Project No. R9804) from the Hong Kong University of Science and Technology (HKUST) and the Early Career Scheme (ECS) from the Research Grants Council (RGC) of Hong Kong (Project Number: 26307123).

## Author contributions

J.L. and Y.Q. conceived and designed the experiments. J.L., J.O., T.L., and F.L. performed experiments. H.S. and I.W. performed the single-crystal XRD analysis. J.L., J.O., and Y.Q. wrote the manuscript. Y.Q. directed the research.

## Competing interests

The authors declare no competing interests.
