## [Peer Review File · Nature Communications]

Metal-Organic Framework Boosts Heterogeneous Electron Donor–Acceptor CatalysisREVIEWER COMMENTS

Reviewer #1 (Remarks to the Author):

In this study, Quan's group fabricated a Zr-PZDB MOF containing Zr₆ clusters and PZDB ligands, which can serve as an efficient heterogeneous catalyst as an electron-donor during electron donor–acceptor catalysis. The outstanding catalytic activity and stability of PZDB ligands existing as the MOF state is noteworthy, and the authors strived to prove this finding in this study. This work is of significance to the photocatalysis field, in particular the field involving diaryl dihydrophenazines. The authors performed enough catalytic experiments and the related characterizations, and I believe the data ought to be credible. Anyway, there are some important issues that need further discussion and elucidation.

1. The authors claimed that 'the high-local concentration of dihydrophenazine active centers and confinement effect in Zr-PZDB are believed to promote the EDA interaction'. I approve the reason of high-local concentration of active sites, but I am confused by the reason of confinement effect. In fact, the authors did not give powerful experimental evidences to verify the 'confinement effect'. The authors claimed that the shortest distance between two adjacent linkers is measured as $\sim 2 \text{ \AA}$, which ought to be inaccessible for all substrate molecules used in this study.

2. The CO₂ Adsorption-Desorption isotherm to evaluate the specific surface area and pore size distribution is inadmissible, and a N₂ Adsorption-Desorption isotherm is necessary. In fact, the BET based on the CO₂ A-D isotherms is really low for a MOF. I wonder the so-called confinement effect for such material. By the way, in the title of 2.2.4, should it be 'adsorption', but not 'absorption'?

3. I am surprised that the type of base influenced the catalysis so markedly, as shown in Table 1. The authors should discuss and explain this phenomenon more.

4. I suggest the authors employing activated carbons or zeolites to load the PZDB-Me or PZDB-H with high-concentration as catalysts to check whether the MOF state is necessary for the outstanding catalytic activity and stability of PZDB.

5. The authors should optimize the use of acronym. For example, EDA is a generally accepted shorthand for ethylenediamine in chemistry field, and the SET was only defined in the Abstract but not in the main text.

Reviewer #2 (Remarks to the Author):

The authors describe an interesting heterogeneous Zr-PZDB MOF, which shows photocatalytic activity in the Minisci reaction and in various heteroaromatics arylation. Cationic pyridinium or sulfonium reagents are used in the model reactions chosen.

Spectroscopic studies suggest the formation of EDA complexes between the Zr-PZDB and the cationic reagents. Interestingly, the same charge-transfer complexes are not observed when the model PZDB-H systems (not connected to the MOF) are mixed with the cationic reagents, thus suggesting the active participation of the MOF in confining and somehow promoting the formation of the EDA complexes.

While this report presents an interesting concept which is suitable to be presented to the broad readership of Nat. Comm. I have major concerns that should be addressed before this report is considered for publication.

- The model reaction in Table 1 and 2 is known to proceed without irradiation, under very mild conditions and in the absence of any catalyst, although with a different pyridinium counterion (Org. Biomol. Chem., 2020, 18, 1738–1742). Although it is clear from table 1 that, under the specific conditions presented, light is required and the MOF is operating, the model reaction chosen do not prove the usefulness of this innovation, as it is more conveniently performed in the absence of any catalyst.

The authors should identify a different model reaction or convincingly explain what is the benefit of their approach in comparison with the very practical methodology mentioned above for the present reaction.

- The key conceptual innovation of this approach is the intervention of an EDA complex. However, for the catalytic reactions the authors use a 427 nm light source, which is outside of the charge-transfer band (see Fig. 2h). It is difficult to assess if the EDA is actually involved in the chemistry, or if instead this is a simple photo-induced electron transfer from the photoexcited Zr-PZDB. This is a possible scenario especially considering that EDAs are usually weak complexes with low association constant.

To prove the active intervention of the EDA complex in the chemistry, the authors should perform experiments with monochromatic light sources (using band-pass filters). Various regions of the charge-transfer band can be excited, monitoring the impact in the catalytic activity. See J. Am. Chem. Soc. 2016, 138, 8019–8030 for an example of the use of monochromatic light to elucidate EDA complex photoactivity.

- The authors propose a close catalytic cycle over a radical chain propagation on the basis of light on/off experiments. Radicals are reactive species with short lifetime which often require continuous initiation to operate. Thus, while for the first reaction a radical chain propagation is clearly operating, for the second reaction (arylation using sulfonium salts) the light on/off experiments do not exclude a radical chain propagation (for a detailed study of the incorrectness of light on/off experiments, see: Chem. Sci., 2015, 6, 5426–5434). In order to characterize a possible radical chain propagation mechanism, the measurement of the reaction quantum yield using a chemical actinometer is required.

Reviewer #3 (Remarks to the Author):

This article by Quan and coworkers describes the design and the use of MOF Zr-PZDB consisting of Zr₆-clusters and PZDB linked as heterogeneous photocatalysts for the Minisci reaction and the arylation of pyrroles. The assumed driving force of both reactions relied on forming electron-donor-acceptor complexes between Zr-PZDB and pyridinium salt or aryl sulfonium salt. The formation of those complexes has been elucidated spectroscopically (UV-vis). The scope of both reactions is relatively convincing and some functionalities are well tolerated under the used conditions.

The manuscript is well-written and the supporting information contains all synthetic and analytic data.

I think this article is publishable but not in its current form. More work (see below) is needed before the submission of a revised version to Nat. Commun.

1) It is for instance very difficult to see the originality of this approach to previous methods reported by Sungwoo Hong, where C2 or C4 functionalized pyridine could be obtained in high yields and excellent regioselectivity. Surprisingly, the regioselectivity in the Minisci reaction is not addressed at all here. The authors should investigate specifically this problem.

2) The same remark holds for the arylation of pyrroles. The scope should be extended to indoles and related structures.

3) The formation of the EDA complex between the photocatalyst and the pyridinium ion is not unprecedented as Quint et al reported the formation of such a complex when Eosin Y is used as a photocatalyst (please see: J. Am. Chem. Soc. 2016, 138, 7436). This work should be quoted and discussed.

4) To be honest, I'm not convinced with the use of the term "EDA catalysis". Catalysts should not be consumed during the reaction. However, both pyridinium and sulfonium are consumed after a single electron transfer, and only MOF-DC is regenerated. Thus, in my point of view, the latter should be the catalyst and not the EDA complex.

5) It is unclear why some references, which are directly related to the present work (for instance Org. Lett. 2020, 22, 19, 7671) are quoted in the supporting information but not in the main text?

Reviewer 1: In this study, Quan’s group fabricated a Zr-PZDB MOF containing Zr₆ clusters and PZDB ligands, which can serve as an efficient heterogeneous catalyst as an electron-donor during electron donor–acceptor catalysis. The outstanding catalytic activity and stability of PZDB ligands existing as the MOF state is noteworthy, and the authors strived to prove this finding in this study. This work is of significance to the photocatalysis field, in particular the field involving diaryl dihydrophenazines. The authors performed enough catalytic experiments and the related characterizations, and I believe the data ought to be credible. Anyway, there are some important issues that need further discussion and elucidation.

ANS: We thank the reviewer for the recognition of our research.

1. The authors claimed that ‘the high-local concentration of dihydrophenazine active centers and confinement effect in Zr-PZDB are believed to promote the EDA interaction’. I approve the reason of high-local concentration of active sites, but I am confused by the reason of confinement effect. In fact, the authors did not give powerful experimental evidences to verify the ‘confinement effect’. The authors claimed that the shortest distance between two adjacent linkers is measured as ~ 2 Å, which ought to be inaccessible for all substrate molecules used in this study.

ANS: We thank this reviewer for the insightful question. In fact, the **shortest** distance between two adjacent linkers is measured as ~ 2 Å (Figures 1a and S5 in the SI). However, the PZDB linker is found to rotate along the backbone (Figures 1b and S4a in the SI), which provides chances for the substrates to diffuse into the pores or channels of MOF. We preliminarily thought that after getting into pores or channels, the substrates might face difficulties in getting out. We currently do not have enough experimental evidence to verify our hypothesis. Therefore, to avoid misunderstanding, the statement of “confinement effect” has been removed from the manuscript.

Figure 1. (a) Single crystal structure showing the shortest distance between two adjacent linkers. (b) Single crystal structure showing the rotation of PZDB linker.

To further prove the considerable interaction between Zr-PZDB and pyridinium salt **1a**, an additional control experiment was designed and performed (Scheme 1). A mixture of **1a** (11.8

mg, 0.05 mmol), **Zr-PZDB** (26 mg, 0.05 mmol, 1 equiv based on the linker), and CH₃CN (1 mL) was stirred at room temperature in dark for 12 h. Then, the MOF solid was separated via centrifugation, and washed with CH₃CN (2 mL * 3). All the CH₃CN portions were combined and then dried under vacuum. The residue was subjected to ¹H NMR analysis to identify the amount of **1a** being remained in CH₃CN solution, which was measured as only 5%. In other words, 95% of **1a** was adsorbed by **Zr-PZDB**, indicating their strong interaction. This result has been added to SI as Section 4.8.

Scheme 1. Adsorption of **1a** by **Zr-PZDB**.

2. The CO₂ Adsorption-Desorption isotherm to evaluate the specific surface area and pore size distribution is inadmissible, and a N₂ Adsorption-Desorption isotherm is necessary. In fact, the BET based on the CO₂ A-D isotherms is really low for a MOF. I wonder the so-called confinement effect for such material. By the way, in the title of 2.2.4, should it be ‘adsorption’, but not ‘absorption’?

ANS: As per suggestion, the N₂ Adsorption-Desorption isotherm was achieved (Figure 2 and S9 in the SI). The surface area was measured as 715 m²/g with the calculated main pore windows of ~ 1.3 nm. In addition, the typo has been corrected in the revised SI.

Figure 2. N₂ Adsorption-Desorption isotherms (left) and pore size distribution profiles (right) of **Zr-PZDB**.

3. I am surprised that the type of base influenced the catalysis so markedly, as shown in Table 1. The authors should discuss and explain this phenomenon more.

ANS: We thank this reviewer for the insightful question. The superior performance of NaHCO₃ is owing to its relatively better solubility compared with that of other common bases, such as Na₂CO₃, K₂CO₃, and Cs₂CO₃. The poor performance of NEt₃ may be attributed to its potential to quench methoxy radical via hydrogen atom transfer reaction (Scheme 2). The corresponding explanations have been involved in the revised manuscript.

Scheme 2. Possible reaction between methoxy radical and NEt₃.

4. I suggest the authors employing activated carbons or zeolites to load the PZDB-Me or PZDB-H with high-concentration as catalysts to check whether the MOF state is necessary for the outstanding catalytic activity and stability of PZDB.

ANS: We thank this reviewer for the constructive suggestion. PZDB-H was loaded into activated carbons. However, the new PZDB-H@C catalyst provided a decreased yield of 15% or 36% for **3a** or **6a**, respectively. These results have been added to Table 1 in the revised manuscript and Section 4.7 in SI.

5. The authors should optimize the use of acronym. For example, EDA is a generally accepted shorthand for ethylenediamine in chemistry field, and the SET was only defined in the Abstract but not in the main text.

ANS: We thank this reviewer for the kind reminder. We have defined “EDA” and “SET” in the revised main text.

Reviewer 2: The authors describe an interesting heterogeneous Zr-PZDB MOF, which shows photocatalytic activity in the Minisci reaction and in various heteroaromatics arylation. Cationic pyridinium or sulfonium reagents are used in the model reactions chosen. Spectroscopic studies suggest the formation of EDA complexes between the Zr-PZDB and the cationic reagents. Interestingly, the same charge-transfer complexes are not observed when the model PZDB-H systems (not connected to the MOF) are mixed with the cationic reagents, thus suggesting the active participation of the MOF in confining and somehow promoting the formation of the EDA complexes. While this report presents an interesting concept which is suitable to be presented to the broad readership of Nat. Comm. I have major concerns that should be addressed before this report is considered for publication.

ANS: We thank the reviewer for the recognition of our research.

The model reaction in Table 1 and 2 is known to proceed without irradiation, under very mild conditions and in the absence of any catalyst, although with a different pyridinium counterion (*Org. Biomol. Chem.*, 2020, 18, 1738–1742). Although it is clear from table 1 that, under the specific conditions presented, light is required and the MOF is operating, the model reaction chosen do not prove the usefulness of this innovation, as it is more conveniently performed in the absence of any catalyst. The authors should identify a different model reaction or convincingly explain what is the benefit of their approach in comparison with the very practical methodology mentioned above for the present reaction.

ANS: We thank this reviewer for the constructive suggestion. The method reported in the corresponding paper (*Org. Biomol. Chem.*, 2020, 18, 1738–1742) is relatively simple however very limited. The initiation step is the deprotonation of benzyl C-H bond in the corresponding quinolinium/pyridinium salt by K_3PO_4 . In other words, substrates without benzyl C-H bonds are not tolerated by this method. To further illustrate the difference between our methodology and the reported one, additional control experiments were performed (Scheme 3). Several pyridinium salts without benzyl C-H bonds were tested under the reported reaction conditions. Only trace amounts of or no target products was detected. In comparison, the same substrates worked well to provide isolated yields of 42% to quantitative for target products under our MOF-catalytic conditions. The related publication has been cited as ref. 79 in the revised manuscript.

Scheme 3. Substrate scope comparison between our method and the reported one.

The key conceptual innovation of this approach is the intervention of an EDA complex. However, for the catalytic reactions the authors use a 427 nm light source, which is outside of the charge-transfer band (see Fig. 2h). It is difficult to assess if the EDA is actually involved in the chemistry, or if instead this is a simple photo-induced electron transfer from the photoexcited Zr-PZDB. This is a possible scenario especially considering that EDAs are usually weak complexes with low association constant. To prove the active intervention of the EDA complex in the chemistry, the authors should perform experiments with monochromatic light sources (using band-pass filters). Various regions of the charge-transfer band can be excited, monitoring the impact in the catalytic activity. See *J. Am. Chem. Soc.* 2016, 138, 8019–8030 for an example of the use of monochromatic light to elucidate EDA complex photoactivity.

ANS: We thank this reviewer for the constructive suggestion. We need to clarify that our LED lamp (Kessil PR160L-427) covers lights with wavelength from 390 to 470 nm (Figure 3). On the other hand, the significantly enhanced absorption peak possibly originated from the EDA interaction ranges from 330 to 430 nm (Figure 2h in the manuscript). In fact, the two bands overlap each other. To further illustrate the role of EDA interaction, a 420 nm band filter, which blocks lights with wavelength smaller than 420 nm, was used with Kessil PR160L-427 lamp. No product **3a** was detected (Scheme 4) with **1a** being intact (Figure 4). The utilization of Kessil PR160L-390 lamp (370-420 nm) provided a yield of ~50% for **3a**, with the detection of ~50% 4-methylpyridine (Scheme 4 and Figure 5). The addition of a 380 nm band filter increased the yield to ~80% (Scheme 4 and Figure 6). These results indicate (1) light with wavelength larger than 420 nm cannot induce the reaction; (2) light less than 380 nm in wavelength might decrease the selectivity for **3a**; (3) the likely involvement of EDA interaction in this photoinduced transformation. Information of LED lamps and new control experiments have been added to the revised SI (Section 4.6). The related paper has also been cited as ref. 72.

PRODUCT SPECTRUMS

Figure 3. Spectrums of Kessil PR160L lamps.

Scheme 4. Effects of light source on the reaction of pyridinium salt **1a** with THF.

Figure 4. ^1H NMR spectrum of crude mixture using PR160L-427 lamp with a 420 nm filter.

Figure 5. ^1H NMR spectrum of crude mixture using PR160L-390 lamp.

Figure 6. ^1H NMR spectrum of crude mixture using PR160L-390 lamp with a 380 nm filter.

The authors propose a close catalytic cycle over a radical chain propagation on the basis of light on/off experiments. Radicals are reactive species with short lifetime which often require continuous initiation to operate. Thus, while for the first reaction a radical chain propagation is clearly operating, for the second reaction (arylation using sulfonium salts) the light on/off experiments do not exclude a radical chain propagation (for a detailed study of the incorrectness of light on/off experiments, see: *Chem. Sci.*, 2015, 6, 5426–5434). In order to characterize a possible radical chain propagation mechanism, the measurement of the reaction quantum yield using a chemical actinometer is required.

ANS: We thank this reviewer for the constructive suggestion. According to the corresponding paper (*Chem. Sci.* 2015, 6, 5426–5434), the quantum yield of pyrrole arylation was measured as 0.003 (Schemes 5 and Section 4.5 in the SI), negating the radical chain pathway. The related paper has been cited as ref. 74.

Scheme 5. Measurement of quantum yield for pyrrole arylation.

Reviewer 3: This article by Quan and coworkers describes the design and the use of MOF Zr-PZDB consisting of Zr₆-clusters and PZDB linked as heterogeneous photocatalysts for the Minisci reaction and the arylation of pyrroles. The assumed driving force of both reactions relied on forming electron-donor-acceptor complexes between Zr-PZDB and pyridinium salt or aryl sulfonium salt. The formation of those complexes has been elucidated spectroscopically (UV-vis). The scope of both reactions is relatively convincing and some functionalities are well tolerated under the used conditions. The manuscript is well-written and the supporting information contains all synthetic and analytic data. I think this article is publishable but not in its current form. More work (see below) is needed before the submission of a revised version to *Nat. Commun.*

ANS: We thank the reviewer for the recognition of our research.

1) It is for instance very difficult to see the originality of this approach to previous methods reported by Sungwoo Hong, where C2 or C4 functionalized pyridine could be obtained in high yields and excellent regioselectivity. Surprisingly, the regioselectivity in the Minisci reaction is not addressed at all here. The authors should investigate specifically this problem.

ANS: We thank this reviewer for the insightful question. According to Prof. Hong's seminal research (*Acc. Chem. Res.* **2022**, *55*, 3043–3056), regioselectivity is highly related to the steric and electronic nature of the pyridinium salt. To investigate the influence of our heterogeneous MOF-catalytic system on regioselectivity, several *N*-methoxypyridinium salts were examined (Scheme 6). For the cross-coupling with THF, the C₂ to C₄ ratio ranges from 1.2 to 2.9. Noteworthily, relatively bulky substituents on C₃ position sterically protect the adjacent C₂ and C₄ positions, resulting in very good C₆ selectivity. The coupling reaction with methanol was also carried out to compare with the results of Prof. Lakhdar's pioneering research (*ACS Catal.* **2020**, *10*, 13710–13717). As shown in Scheme 6, C₂ to C₄ selectivity was observed in the same order of magnitude. These new results have been added to the revised manuscript.

Scheme 6. Investigation on the regioselectivity of MOF catalysis. [#] Results from Prof. Lakhdar's pioneering research.

2) The same remark holds for the arylation of pyrroles. the scope should be extended to indoles and related structures.

ANS: We thank this reviewer for the insightful question. Other heterocycles were examined accordingly (Scheme 7). Indoles with electron withdrawing functional groups also worked well. Thiophene derivatives with different substituents, including bromo, ester, and methoxyl, reacted with the sulfonium salt smoothly to deliver the coupling products in 52-82% isolated yields. Moreover, the substrate scope was further extended to include furan, giving the target product in 91% yield. These results have been added to the revised manuscript.

Scheme 7. Examination of other heterocycles.

3) The formation of the EDA complex between the photocatalyst and the pyridinium ion is not unprecedented as Quint et al reported the formation of such a complex when Eosin Y is used as a photocatalyst (please see: J. Am. Chem. Soc. 2016, 138, 7436). This work should be quoted and discussed.

ANS: As per suggestion, the related paper has been cited as ref. 71. A brief introduction of the pioneering research has also been added to the revised manuscript.

4) To be honest, I'm not convinced with the use of the term "EDA catalysis". Catalysts should not be consumed during the reaction. However, both pyridinium and sulfonium are consumed after a single electron transfer, and only MOF-DC is regenerated. Thus, in my point of view, the latter should be the catalyst and not the EDA complex.

ANS: We understand this reviewer's concern. Currently, the definition of EDA catalysis involves the use of an electron donor or acceptor as the catalyst. To avoid misunderstanding, the statement "EDA catalysis" has been changed to "EDA photoactivation" in the revised manuscript.

5) It is unclear why some references, which are directly related to the present work (for instance Org. Lett. 2020, 22, 19, 7671) are quoted in the supporting information but not in the main text?

ANS: As per suggestion, the related paper has been cited as ref. 78 in the revised manuscript.

REVIEWERS' COMMENTS

Reviewer #1 (Remarks to the Author):

The authors have made revisions according to the comments. The revised manuscript is recommended for publication.

Reviewer #2 (Remarks to the Author):

The authors have carried out additional experiments and have provided a convincing response to my initial concerns. This article is now ready to be published in nature communications.